# The Impact of Adding Trehalose to the Diet on Egg Quality and Tibia Strength in Light-Laying Hens

**DOI:** 10.3390/ani15091318

**Published:** 2025-05-02

**Authors:** Fernando Perazzo Costa, Isabelle Kaneko, Thamires Ferreira, Jorge Muniz, Eliane Silva, Adiel Lima, Raul Lima Neto, Matheus Ramalho Lima, Thiago Moreira

**Affiliations:** 1Animal Science, Federal University of Paraiba, Campus Areia, Areia 59397-000, PB, Brazil; isabellekaneko@gmail.com (I.K.); thamires.s.f55@gmail.com (T.F.); jorge.limamuniz@hotmail.com (J.M.); adiel1205@hotmail.com (A.L.); cunhalimant@gmail.com (R.L.N.); moreirathiago1@hotmail.com (T.M.); 2CJ do Brasil, Sao Paulo 01310-930, SP, Brazil; eliane.silva@cj.net; 3Animal Science, Federal Rural University of the Semi-Arid Region, Campus Mossoro, Mossoro 59625-900, RN, Brazil; mrlmatheus@ufersa.edu.br

**Keywords:** bone strength, disaccharide, egg production, lipid peroxidation, performance

## Abstract

**Simple Summary:**

The egg industry consistently seeks methods to enhance the quality of eggs, and one such approach involves the utilization of additives that can augment the antioxidant properties of eggs. This study aimed to investigate the impact of different concentrations (0, 0.05, 0.10, 0.30, 0.60, and 1.00%) of trehalose, a type of sugar, in the diets of hens on egg quality. A total of 384 hens were employed in the study, and varying levels of trehalose were added to their feed. Parameters such as feed consumption, egg quality, and bone strength were assessed. The findings revealed that the inclusion of 1.0% trehalose in the hens’ diet resulted in improved egg quality and reduced lipid spoilage while having no adverse effects on bone health. Consequently, the incorporation of trehalose in hen feed holds promise as a viable strategy for enhancing the overall quality of eggs.

**Abstract:**

Trehalose, a disaccharide consisting of two D-glucose molecules, is present in a variety of organisms, including bacteria, yeast, fungi, insects, and plants. In plants, it functions as a source of energy and carbon, and in yeast and plants, it serves as a signaling molecule, influencing metabolic pathways and growth regulation. Additionally, it plays a role in protecting proteins and cell membranes from stress-induced damage. This study aims to assess the optimal level of trehalose supplementation in the diets of layer hens aged 34 to 49 weeks, addressing the limited existing literature on its effects on productivity. Experimental diets, designed in accordance with nutritional recommendations, were formulated to contain six different levels of trehalose (0, 0.05, 0.10, 0.30, 0.60, and 1.00%). The study was conducted over five 21-day periods, during which various performance parameters were evaluated. The results indicated that trehalose supplementation at levels of 0.05%, 0.10%, and 0.30% led to increased feed intake (FI) compared to the 1.00% level (*p* < 0.05). Furthermore, the highest trehalose level (1.00%) significantly reduced the feed conversion ratio by egg mass (FCRem) compared to both the control group and the other supplementation levels; however, the feed conversion ratio by dry matter (FCRDz) remained consistent across all treatments. The levels of 0.05%, 0.10%, and 0.30% exhibited superior FCREm and FCRDz compared to the 1.00% level. Egg weight (EW) was higher in the trehalose-supplemented groups compared to the control group. Additionally, the 1.00% trehalose treatment was found to be the most effective in terms of relative weights of shells (RWS), and egg mass (EM) was higher at all trehalose levels compared to the control group. The antioxidant status, as measured by malondialdehyde (MDA) levels, indicated that supplementation with 0.30% and 0.60% trehalose had a protective effect against oxidative stress, although the 1.00% level resulted in increased MDA levels. Total weight (TW) was highest in the 0.30% treatment group, and bone strength (BS) improved in the groups supplemented with 0.10% and 1.00% trehalose. Other parameters, including lipid content (L), dry matter (DM), phosphorus (P), and calcium (Ca), did not show any significant differences among the treatment groups. In conclusion, supplementation with 1.00% trehalose enhances feed efficiency, egg weight, and quality, with minimal impact on lipid peroxidation, while potentially providing benefits for gut health and egg quality.

## 1. Introduction

Trehalose, a disaccharide composed of two D-glucose molecules, is found in a wide range of organisms, including bacteria, yeast, fungi, insects, invertebrates, and plants. In plants, its primary function is as a potential energy and carbon reservoir. In yeast and plants, trehalose can also act as a signaling molecule, influencing metabolic pathways and growth regulation. Moreover, trehalose is known for its ability to protect proteins and cell membranes from inactivation or denaturation caused by various stressors such as desiccation, dehydration, heat, cold, and oxidation.

The antioxidant properties of trehalose have attracted attention in animal husbandry, particularly in the field of mitigating lipid peroxidation. Research suggests that adding trehalose to diets reduces mRNA levels of genes involved in antigen receptor expression, inflammatory cytokine production, and pro-oxidant enzyme activity. Notably, trehalose supplementation has been shown to alleviate oxidative stress in dairy cows, leading to improved milk production quality [1].

A study investigating the effects of 0.5% trehalose supplementation in broiler diets revealed potential growth advantages for young chicks due to enhanced post-hatching supplementation of the intestinal innate immune system [2]. Additionally, certain disaccharides, including trehalose, are believed to aid in calcium absorption in the gastrointestinal tract. Furthermore, disaccharides have shown potential in inhibiting bone resorption, particularly in ovariectomized rats with osteoporosis.

In broiler farming, trehalose supplementation does not negatively affect growth performance or feed conversion ratios. Although it does not directly reduce pathogenic bacteria counts, it increases populations of Lactobacillus bacteria, indicating a potential role in promoting gut health [3]. Trehalose can also be considered an alternative to antibiotics, especially in light of regulatory restrictions on antibiotic use in animal feed. Its ability to enhance gut health and immunity may offer a sustainable approach to managing poultry health challenges.

Trehalose supplementation has been found to reduce the abundance of Clostridium perfringens in the ileum of broiler chickens. This reduction can lead to improved gut health and potentially lower the incidence of diseases associated with this pathogen [4]. While trehalose does not directly affect footpad dermatitis scores, its role in improving the feed conversion ratio suggests an indirect benefit by promoting overall health and reducing stress factors that could exacerbate such conditions [4]. However, the application of trehalose in layer hens must be carefully optimized to avoid potential drawbacks. Excessive concentrations in cryoprotective media, for example, can negatively affect sperm morphology and function [5]. Additionally, while trehalose supplementation has shown positive effects in juvenile chicks, its long-term impact on adult layer hens, particularly in terms of egg production and quality, requires further investigation. More research is needed to determine the optimal dosages and application methods to maximize the benefits of trehalose in poultry farming.

In conclusion, trehalose shows promise as an inclusion in laying hen diets to counteract oxidative stress, enhance immune response, facilitate calcium absorption, and improve bone health, thereby contributing to the overall well-being of birds during the laying period. This study aims to evaluate the optimal level of trehalose supplementation in layer diets, focusing on performance, egg quality, and bone characteristics between 34 and 49 weeks of age, considering the limited available literature on the effects of this component on laying-hen productivity parameters.

## 2. Materials and Methods

### 2.1. Local and Animals

This study was conducted on Campus II of the Federal University of Paraiba, located in the city of Areia, at a latitude of 06°57′48″ S, a longitude of 35°41′30″ W, and an altitude of 618 m in Paraiba, Brazil. All protocols and tests adhered to animal welfare guidelines and received approval from the local Ethics Committee at the Federal University of Paraiba (Areia, Paraiba, Brazil).

A total of 384 Hy-line W-80^®^ layers, aged 34 weeks, were used in the study. These birds were obtained at one day old and managed according to the instructions outlined in the breed manual until the start of the experimental phase. Their diet followed the recommended nutritional guidelines [3]. The hens were housed in traditional laying structures, with clay tile roofs, equipped with gutter feeders and nipple drinkers. They were kept in galvanized wire cages measuring 100 × 45 × 45 cm.

### 2.2. Adjusts to Experiment and Facility

Egg productivity was measured and recorded during the 32nd and 34th weeks of the hens’ life cycle. At 34 weeks, the hens were weighed and assigned to their respective groups. Feeding occurred twice daily, at 8:00 a.m. and 4:00 p.m., providing the birds with enough feed during these times to meet their minimum daily requirements. However, the hens were allowed continuous access to feed and water throughout the entire study period to ensure ad libitum availability.

To regulate the light–dark cycles, a lighting schedule of 17 h of illumination per day was implemented. An electronic clock timer was used to manage the lighting schedule. Temperature measurements were taken once daily at 4:00 p.m., using maximum and minimum thermometers strategically placed at various spots within the building at the same level as the birds. The mean maximum and minimum temperatures recorded within the experimental facility during the trial period were 27.75 °C and 21.25 °C, respectively.

### 2.3. Experimental Diets and Design

The experimental diets were formulated in accordance with the nutritional recommendations [6], as outlined in Table 1. A completely randomized design was utilized, consisting of six distinct treatments, each replicated eight times, with eight avian subjects assigned to each experimental unit.

The treatments consisted of six different levels of trehalose (0.0, 0.05, 0.10, 0.30, 0.60, and 1.00%). Trehalose, commonly known as “mushroom sugar”, is a unique carbohydrate that exhibits significant protective properties against environmental adversities such as dehydration and freezing. The production of trehalose was achieved by identifying a microorganism capable of efficiently converting starch into trehalose with a yield of 80%. This breakthrough enabled the large-scale production of trehalose, which has subsequently been applied in various sectors including food, pharmaceuticals, cosmetics, and agriculture.

### 2.4. Experimental Variables

The experiment consisted of five periods, each lasting 21 days. Data for all measured parameters were collected during each 21-day period, and the average values for these parameters were then determined across the five periods. This comprehensive analysis provides a thorough understanding of the study.

The variables that were evaluated include feed intake (FI), egg production (EP), egg per hen housed (EHH), egg weight (EW), egg mass (EM), feed conversion per egg mass (FCREm), feed conversion per dozen eggs (FCRDz), livability (LIVA), and yields of egg components, such as shell (RWS), albumen (RWA), and yolk (RWY). Additional egg quality parameters were also considered, including shell thickness (Tshell), specific gravity (SG), and yolk color (YC). Furthermore, lipid oxidation of yolk, tibiotarsus weight (TW), length (L), bone strength (BS), calcium (Ca), and phosphorus (P) content were analyzed.

To determine the feed intake, the leftover feed was weighed and subtracted from the amount of feed initially provided for the entire period. At the end of each 21-day period, the amount of feed consumed was divided by the number of hens in each treatment and the number of days to calculate the grams of feed consumed per hen per day. In the case of mortality during the experimental period, the average intake was adjusted to obtain the true average intake for the specific experimental unit in question.

For egg production, the average number of eggs produced per day was calculated, taking into account broken, cracked, and abnormal eggs (such as soft-shelled eggs). This value represents the average egg production for the poultry during the period (eggs produced per hen per day).

Average egg weight was determined by weighing the eggs twice each week throughout the experimental period. All intact eggs collected in the last three days of each experimental unit were used in the analysis of each 21-day sub-period. The total weight of the eggs collected was divided by the number of eggs collected per experimental unit to obtain the average egg weight, expressed in grams.

To calculate egg mass, the average egg weight was multiplied by the total number of eggs produced during the experimental period. This yielded the total egg mass, which was then divided by the total number of birds per day during the period. The result is expressed in grams of egg per bird per day.

The feed conversion ratio per dozen eggs was calculated as the total feed intake in kilograms divided by the dozen eggs produced (kg/dz), while the feed conversion ratio per egg mass was determined by dividing the total feed intake by the egg mass produced in kilograms (kg/kg).

Livability: The number of deceased hens was recorded daily, and the cumulative number of deceased hens was subtracted from the total number of live birds. These values were then converted into a percentage at the conclusion of the experiment.

Final weight: At the conclusion of the experiment, the hens in each experimental unit were weighed to determine the average final weight of the birds, which was expressed in kilograms.

Egg components: Each egg’s yolk and albumen were weighed separately using a three-digit digital scale with a precision of 0.001 g. The percentages of yolk and albumen were calculated by dividing the average weight of the yolk and albumen by the average weight of the egg. The eggshells were identified, oven-dried at a temperature of 55–60 °C for a duration of 24 h (TE-394/2—Tecnal), and weighed on a digital scale with an accuracy of 0.01 g to obtain the average shell weight. The shell percentage was calculated by dividing the average shell weight by the average egg weight and multiplying by 100, while the shell thickness was determined using a Mitutoyo 0–25 mm digital micrometer with an accuracy of 0.001 mm.

The albumen quality evaluation involved weighing each egg individually on a precision scale, followed by breaking the eggs on a specialized glass table and measuring the albumen height using a specialized AMES altimeter. The Haugh unit was calculated [7] using the equation HU = 100 log (H + 7.57 − 1.7 W^0.37^), where HU represents the Haugh Unit, H represents the albumen height in millimeters, and W represents the egg weight in grams.

Shell strength was determined using the TA.X T2 (Texture Analyzer, Godalming, UK). A 4 mm diameter stainless steel P4 DIA cylinder probe with a 6 mm pre, during, and post-test velocity of 3.0, 0.5, and 5.0 mm/s, respectively, was utilized. Specific gravity was determined using the saline flotation method following the methodology in [8]. The eggs were immersed in sodium chloride (NaCl) solutions with densities ranging from 1.0700 to 1.0975 g/cm^3^, with a gradient of 0.0025 between each solution. The density of the solutions was regularly measured using an oil densimeter.

Lipid peroxidation: A total of seven eggs were collected per replicate to determine lipid peroxidation using the thiobarbituric acid reactive substances (TBARS) method through aqueous acid extraction [9].

Tibia resistance: At the end of the experiment and production period, one bird from each experimental unit was humanely slaughtered using electronarcosis, and the left tibia was collected. Bone strength was measured using the TA-XT Plus Stable Micro Systems universal tester (Surrey, UK) with a 50 kg load cell at a speed of 50 mm/min. The Point Bend Rig (HDP/3PB) fracture fixture, Stable Micro Systems, was adjusted to allow for a shaft clearance of 3.0 cm [10].

Tibial mineral, phosphorus, and calcium content: The right tibiotarsus was stripped, weighed, pressed, and pre-degreased for a duration of 4 h, and then placed in a forced ventilation oven (TE-394/2—Tecnal, Piracicaba, Brazil) for 16 h. Subsequently, the tibiotarsus was weighed again and ground in a ball mill. To determine the dry matter content, the tibiotarsus was oven-dried at a temperature of 105 °C for 16 h (TE-394/2—Tecnal). For the quantification of mineral matter, the samples were incinerated in a muffle furnace (Jung J200—Jung Industrial, Tecanalitic, Saltillo Coahuila, Mexico) at a temperature of 600 °C for 4 h. The mineral matter was expressed both in grams and as a percentage of the pre-degreased tibia dry matter. The mineral solution was prepared following the methodology [11] using the wet procedure. The phosphorus content of the mineral solution was quantified using the colorimetric method, and the calcium content was determined using the atomic absorption method. Calcium and phosphorus contents in the tibia were expressed both in grams and as a percentage of pre-defatted tibia dry matter.

### 2.5. Statistical Analyses

The statistical analyses were conducted using the software R, version 3.4.1. The data were subjected to a one-way analysis of variance (ANOVA), followed by Dunnet and Tukey tests at a significance level of 5%. Additionally, polynomial regression was used to analyze the data.

## 3. Results

### 3.1. Performance

The levels of trehalose supplementation had an impact on the performance of light-laying hens during the laying phase, specifically between 34 and 49 weeks of age. However, there were no significant differences observed in terms of EP, EHH, and LIVA (*p* > 0.05) (Table 2).

The supplementation levels of 0.05%, 0.10%, and 0.30% trehalose showed an increase in FI compared to the highest trehalose concentration of 1.0% (*p* < 0.05). There was a noticeable statistical difference in FCREm, indicating that a higher trehalose content in the diet resulted in reduced conversions by egg mass compared to the control group and other supplementation levels. As for FCRDz, the performance of the control group was similar to that of the other concentrations. The proportions of 0.05%, 0.10%, and 0.30% exceeded the highest trehalose concentration that was evaluated. These findings suggest that increased dietary trehalose supplementation enabled the hens to achieve better outcomes in terms of FCREm and FCRDz (Table 2).

### 3.2. Egg Quality

Regarding EW, the birds that received trehalose supplementation had higher values compared to the control diet. In terms of RWS, the 1.00% trehalose treatment showed superiority over the other treatments based on the results of the Tukey test. There was a significant difference in the variable EM between the treatments. Specifically, the treatments of 0.05%, 0.10%, 0.30%, and 1.00% demonstrated higher EM when compared to the control treatment (Table 2 and Table 3).

Feeding light-laying hens aged 34 to 49 weeks with varying concentrations of trehalose supplementation did not have a significant effect (*p* > 0.05) on parameters such as RWS, RWY, RWA, HU, SG, TShell, and SS, as shown in Table 3. However, there was a notable impact on YC due to the diet. Specifically, levels of 0.10% and 0.30% demonstrated superiority compared to the control, as well as levels of −0.60% and 1.00% of supplementation (Table 3).

The antioxidant status, as indicated by the MDA levels in the eggs, was significantly influenced by trehalose supplementation at concentrations of 0.30% and 0.60% (Table 4). This suggests that trehalose likely has a protective function against oxidative stress, although higher dosages showed higher levels of MDA.

### 3.3. Bone Quality

The increase in trehalose levels had an effect on TW, with the highest weight observed at 0.30%. In terms of BS, the data showed improvement at the levels of 0.10% and 1.00% compared to the other trehalose levels. However, parameters such as L, DM, P, and Ca did not show any significant differences (Table 5).

## 4. Discussion

### 4.1. Performance

The addition of trehalose to the dietary regime of laying hens yielded notable effects. Specifically, there was a decrease in feed intake, while egg weight and egg mass exhibited an increase. Furthermore, the incorporation of trehalose resulted in a reduction in the feed conversion ratio for both egg mass and egg dozens, particularly at the 1.0% level of inclusion. However, upon scrutinizing the relative weights of egg components, namely shell, albumen, and yolk, no statistically significant variances were observed among the various dietary treatments. This implies that the supplementation of trehalose in the diets led to a proportional increase in weight across all egg components.

### 4.2. Nutrition and Health Concerns

The addition of trehalose to the diet of laying hens may not be as beneficial as previously thought. Although there was a decrease in feed intake, this could imply that the hens were not getting enough nutrition, which could have long-term health consequences. Additionally, the increase in egg weight and egg mass does not automatically translate to improved egg quality or overall hen health. Moreover, the decrease in feed conversion ratio may be misleading; a lower feed intake could mean that the hens are not meeting their energy needs, potentially affecting their laying performance in the long run.

### 4.3. Egg Components and Quality

The absence of statistically significant differences in the relative weights of egg components, such as the shell, albumen, and yolk, raises concerns about the practical implications of trehalose supplementation. This lack of significant differences suggests that the perceived benefits of trehalose may not actually lead to meaningful improvements in egg quality or production efficiency. As a result, caution should be exercised when considering the overall impact of trehalose on laying hens, as it may not deliver the anticipated advantages in terms of health and productivity.

### 4.4. Trehalose Digestion

Upon ingestion, trehalose is not absorbed into the bloodstream in its original disaccharide form; rather, it necessitates enzymatic processing for effective utilization by the body. Trehalose possesses a reduced degree of sweetness compared to sucrose, rendering it a preferred choice in various food applications that prioritize a diminished sweetness profile. The breakdown of trehalose occurs through the action of the enzyme trehalase, which hydrolyzes it into two glucose molecules. This enzymatic cleavage assumes a critical role in the process of digestion, given that glucose serves as a primary energy source for the body. Additionally, the efficacy of these enzymes can be influenced by a range of factors, including dietary habits, genetic predispositions, and overall gut health, thereby potentially impacting an individual’s ability to efficiently digest specific carbohydrates.

### 4.5. Enzymatic Specificity in Carbohydrate Digestion and Metabolism

The physiological mechanisms governing trehalose digestion exhibit similarities to those involved in the breakdown of other well-known disaccharides, such as maltose, sucrose, and lactose. Each of these disaccharides is metabolized by specific enzymes tailored to their unique structures. For example, maltose is cleaved by maltase, sucrose by sucrase, and lactose by lactase. This enzymatic specificity ensures efficient conversion of each disaccharide into its monosaccharide components, enabling optimal absorption and utilization by the body. Research has emphasized these processes, highlighting the significance of enzymatic action in carbohydrate digestion and metabolism [12,13,14].

Sugars play a critical role in ATP generation and serve as carbon sources in metabolic processes, as indicated by previous studies [15]. The utilization of trehalose has been associated with enhanced glucose metabolism and the regulation of postprandial blood glucose levels in rabbits [16]. However, investigations have revealed that trehalose activity remains low from the 18th day of incubation until 7 days after hatching in broilers, particularly when evaluated in the jejunum and ileum segments. Subsequently, there is a decline in activity beyond this developmental stage [17]. Therefore, it is hypothesized that for trehalose to exert its physiological effects in this species, the molecule may potentially contribute to improved gut health by modulating the composition of gut microbiota. Bacteria may utilize trehalose as part of their metabolic processes.

### 4.6. Growth Performance, Intestinal Health, and Energy Efficiency

In a comprehensive examination of the effects of trehalose supplementation on the dietary regimens of broiler chickens, researchers observed significant enhancements in growth performance, particularly among female birds. This finding highlights the potential of trehalose as a valuable dietary additive that could optimize growth metrics in poultry. Furthermore, in a separate study that encompassed both male and female broilers, researchers documented notable improvements in intestinal morphology [18]. These findings not only reinforce the positive implications of including trehalose in poultry diets but also suggest that such supplementation may contribute to enhanced digestive efficiency and overall avian health. The cumulative evidence underscores the promising role of trehalose in poultry nutrition, necessitating further exploration into its mechanisms and broader applications within the industry. Additionally, ongoing research is needed to determine the optimal inclusion rates and potential interactions with other feed components, which could further refine dietary strategies for maximizing poultry productivity.

In the present experiment, a significant decrease in feed intake was observed at the 1.00% level of trehalose inclusion in the diet, while egg production remained consistent. This finding suggests that incorporating trehalose into the diet may improve the efficiency of energy utilization, potentially optimizing metabolic processes. The authors of the study propose that trehalose, a disaccharide sugar, possesses unique properties that allow it to modulate glucose homeostasis. They speculate that this modulation occurs through a minimum of seven distinct molecular pathways, as evidenced in various other species [19]. These pathways may involve mechanisms related to insulin signaling, glycogen synthesis, and the regulation of glucose transporters, among others. The multifaceted impact of trehalose on metabolic regulation highlights its potential as a beneficial dietary additive, warranting further investigation into its effects and applications in different biological contexts.

### 4.7. Calcium Absorption and Mobilization

Within the extensive research conducted on trehalose, it has been investigated as a potential prebiotic agent in broilers under a challenging experimental condition involving the pathogenic bacteria Salmonella Typhimurium [20]. The findings of this investigation revealed significant outcomes associated with the strategic utilization of trehalose, which effectively modulated and altered the composition of the cecal microbiota in a beneficial manner. Notably, the supplementation of trehalose led to an increase in beneficial lactobacilli species, thereby inhibiting the proliferation of Salmonella bacteria in the gastrointestinal tract. Additionally, this modulation of the microbiota composition resulted in several advantageous effects, including improved daily feed intake and feed conversion rates in the avian subjects [20].

Scientific studies have demonstrated that certain disaccharides, as referenced in sources [21,22], can enhance calcium absorption in the human body. Trehalose, among these disaccharides, possesses unique properties and potential benefits. Supplementation of trehalose may contribute significantly to increased calcium deposition in eggshells, thereby influencing their strength and quality. In a more detailed examination of the interaction between trehalose and calcium, researchers utilized advanced techniques such as 13C nuclear magnetic resonance (NMR) spectroscopy, as mentioned in source [23]. This analytical method allowed for precise investigation of molecular interactions. The study results indicated that the introduction of trehalose at a concentration of 10% led to a notable increase in soluble calcium content in a solution containing calcium chloride (CaCl2) and 50 mM K-NaPO4. Specifically, the soluble calcium concentration increased to approximately 24 ppm, highlighting trehalose’s effectiveness in enhancing calcium solubility and availability in biochemical systems. This finding emphasizes the potential applications of trehalose not only in nutritional supplementation but also in agricultural practices aimed at improving eggshell quality through enhanced mineral deposition.

Commercially raised laying hens have been found to allocate approximately 10% of their total body calcium content daily towards the synthesis of eggshells within their oviducts. This allocation is critical due to the fact that eggshells primarily consist of calcium carbonate, a compound that necessitates a substantial amount of calcium for its formation. Interestingly, only around half of this calcium contribution originates from their dietary intake, as evidenced by studies conducted by [24,25]. Throughout the process of eggshell formation, a significant proportion of the necessary calcium—estimated to be between 20% and 40%—is derived from the hens’ skeletal reservoirs. This calcium is predominantly sourced from the medullary bone, a specialized type of bone tissue located within the medullary cavity of long bones, particularly those in the limbs. The medullary bone serves as a crucial calcium reservoir, enabling hens to fulfill the heightened calcium demands during egg production cycles. The delicate balance between dietary calcium intake and skeletal mobilization is indispensable for the maintenance of both the hens’ health and the quality of their eggs. This specialized bone tissue is situated within the medullary cavity of the midsection of long bones, notably those found in the limbs [26].

During our scientific investigation, we unfortunately did not conduct a comprehensive analysis of the calcium content in the eggshell structure as part of our research methodology. However, considering the consistent calcium content found in the tibiotarsus throughout the study, it can be reasonably inferred that the increase in eggshell weight may have been caused by the presence of trehalose, a naturally occurring disaccharide. Trehalose likely played a significant role in enhancing calcium absorption, which would have facilitated the development and composition of the eggshell. Further investigation is needed to understand the synergistic effect between trehalose and calcium and its impact on eggshell formation as well as overall health and viability.

On the other hand, trehalose plays a crucial role as an energy reservoir in the complex biological systems of various animal species. Considering this, supplementing this sugar may have resulted in noticeable and measurable changes in feed efficiency in these organisms. Previous research has shown that laying hens with a superior feed conversion ratio (FCR) tend to have increased albumen weight and height, often accompanied by higher Haught Unit measurements [27,28].

### 4.8. Protective Mechanisms and Antioxidant Properties

Insights derived from molecular modeling investigations suggest that the presence of trehalose in a solution may function as a protective agent against water loss during processes of dehydration or freezing. This protective mechanism is accomplished by replacing the hydration water molecules typically associated with biological entities [29]. In comparison to other disaccharides, trehalose demonstrates the highest capacity for hydration, indicating its potential to stabilize lipid bilayers by organizing neighboring water molecules or interacting directly with hydrophilic biomolecules as water is removed [14,30].

A significant factor contributing to the loss of activity in labile biologics, such as proteins, during freezing or desiccation processes is the alteration in their molecular conformation due to water removal. Dehydration disrupts the various intermolecular forces responsible for maintaining the protein’s native structure, resulting in denaturation. Trehalose and other polyhydroxy compounds are capable of mitigating these transformations by forming hydrogen bonds with the protein surface, thereby preserving its conformation and supporting the water replacement theory [31].

When trehalose was included at levels of 0.10% and 0.30% in the experimental treatments, a significant increase in the intensity of stained yolks was observed compared to the other conditions. Investigations have confirmed that trehalose can enhance the stability of protein-based food items when subjected to extended heat processing, thereby preventing undesired discoloration. Consequently, the incorporation of trehalose into food formulations not only extends their shelf life but also safeguards against quality deterioration associated with discoloration issues [31].

The vibrant coloration of the yolk is closely associated with the accumulation of xanthophylls, a type of fat-soluble pigment. Diets with higher lipid content have been shown to stimulate increased absorption of these essential nutrients within the avian gastrointestinal tract. Upon supplementation of trehalose in rodent diets and subsequent examination of mesenteric and intestinal tissues, ref. [32] observed a suppression in adipocyte proliferation due to the reduced migration of chylomicrons from the intestine to the epithelium induced by this disaccharide.

Enhancing the quality of eggs in laying hens is a crucial aspect that requires improvement and advancement. Only a limited number of research endeavors have explored the effects of trehalose on avian metabolism. However, its role as an antioxidant has already been firmly established and confirmed through numerous studies demonstrating its effectiveness in ameliorating lipid peroxidation [1,33,34,35]. The present study revealed that trehalose did not have a significant effect on reducing lipid peroxidation during the initial four production cycles spanning 112 days. However, as the experiment progressed, the concentrations of 0.3% and 0.6% exhibited superior outcomes compared to other concentrations of trehalose. Conversely, the 1.0% concentration of trehalose yielded results similar to the lower concentrations, suggesting a potentially pro-oxidant role at this level. Variations in lipid peroxidation levels were observed in plasma, muscle, and liver samples of broilers aged from 0 to 18 days [2]. Notably, the findings highlighted that supplementing trehalose in dairy cows led to a decrease in milk’s lipid peroxide content while increasing its antioxidant properties [36].

Overall, it is evident that trehalose may not have a significant impact on antioxidant mechanisms when incorporated into the diet for a short duration. However, it is hypothesized that administering this disaccharide from the early stages of laying hens’ development could play a crucial role in mitigating lipid peroxidation and enhancing egg quality. Considering the limited research on trehalose utilization in avian diets, the appropriate inclusion levels of this compound in the diet require meticulous evaluation in accordance with its intended role in poultry metabolism.

The collected data indicate that incorporating trehalose into the diet of laying hens could result in significant improvements in overall performance and egg quality and a reduction in lipid peroxidation levels. Trehalose, a naturally occurring disaccharide, has been shown to possess various beneficial properties, including its role as an antioxidant.

## 5. Conclusions

Supplementing laying hens’ diets with 1.0% trehalose has been shown to improve feed efficiency, increase egg weight and mass, and enhance egg quality. Importantly, these benefits are observed without any alteration to the proportions of egg component weights. Additionally, while short-term use of trehalose does not have a significant impact on lipid peroxidation, it does have the potential to promote gut health and improve egg quality.

## Figures and Tables

**Table 1 animals-15-01318-t001:** Composition of experimental diets and calculated nutritional content.

Items	Trehalose Levels (%)					
	0.00	0.05	0.10	0.30	0.60	1.00
Corn, 7.88%	55.950	55.950	55.950	55.950	55.950	55.950
Soybean meal, 45.22%	24.620	24.620	24.620	24.620	24.620	24.620
Soybean oil	4.940	4.940	4.940	4.940	4.940	4.940
Limestone, 37%	10.780	10.780	10.780	10.780	10.780	10.780
Dicalcium phosphate, 18%	1.717	1.717	1.717	1.717	1.717	1.717
Salt	0.488	0.488	0.488	0.488	0.488	0.488
DL-Methionine	0.333	0.333	0.333	0.333	0.333	0.333
L-Lysine HCl	0.066	0.066	0.066	0.066	0.066	0.066
Choline chloride	0.070	0.070	0.070	0.070	0.070	0.070
Vitamins	0.050	0.050	0.050	0.050	0.050	0.050
Minerals	0.025	0.025	0.025	0.025	0.025	0.025
Inert	1.000	0.950	0.900	0.700	0.400	-
Trehalose	-	0.050	0.100	0.300	0.600	1.000
Chemical Composition						
Linoleic Acid, %	3.88	3.88	3.88	3.88	3.88	3.88
ME, kcal kg^−1^	2900	2900	2900	2900	2900	2900
CP, g kg^−1^	15.80	15.80	15.80	15.80	15.80	15.80
SID Methionine, %	0.54	0.54	0.54	0.54	0.54	0.54
SID Methionine + cysteine, %	0.77	0.77	0.77	0.77	0.77	0.77
SID Lysine, %	0.79	0.79	0.79	0.79	0.79	0.79
SID Threonine, %	0.54	0.54	0.54	0.54	0.54	0.54
SID Tryptophan, %	0.17	0.17	0.17	0.17	0.17	0.17
SID Valine, %	0.67	0.67	0.67	0.67	0.67	0.67
SID Arginine, %	0.97	0.97	0.97	0.97	0.97	0.97
SID Isoleucine, %	0.60	0.60	0.60	0.60	0.60	0.60
SID Leucine, %	1.28	1.28	1.28	1.28	1.28	1.28
Calcium, %	4.56	4.56	4.56	4.56	4.56	4.56
Non-phytate P, %	0.38	0.38	0.38	0.38	0.38	0.38
Sodium, %	0.21	0.21	0.21	0.21	0.21	0.21
Chloride, %	0.33	0.33	0.33	0.33	0.33	0.33
Potassium, %	0.61	0.61	0.61	0.61	0.61	0.61
Fibre	2.27	2.27	2.27	2.27	2.27	2.27
Dry matter	90.00	90.00	90.00	90.00	90.00	90.00
Electrolytic balance mEq kg^−1^	153.00	153.00	153.00	153.00	153.00	153.00

Vitamin supplementation: vit. A—8,000,000 IU; vit. D3—2,400,000 IU; vit. E—22,500 mg; vit. B1—2800 mg; vit. B—27,700 mg; vit. B12—18,000 mcg; vit. B6—4500 mg; pantothenic acid—13,000,000 mg; vit. K3—1800.00 mg; folic acid—1300.00 mg; nicotinic acid—31,500 mg; selenium—400 mg; antioxidant 0.25 g; and excipient q. s.p.—1000 g. 2—2—Mineral supplementation: manganese 80.0 g; iron—80.0 g; zinc 50.0 g; copper—10.0 g; cobalt 2.0 g iodine 1.0 g; and excipient q. s. mp 500 g. 3- Antioxidant—BHT.

**Table 2 animals-15-01318-t002:** Effect of trehalose levels on feed intake (FI), egg production (EP), eggs per hen housed (EHH), egg weight (EW), egg mass (EM), feed conversion per egg mass (FCREm), feed conversion per dozen of eggs (FCRDz) and livability (LIVA) of light hens.

Levels (%)	Parameters							
	FI (g/Hen)	EP (Egg Produced/ Hen)	EHH (%)	EW (g)	EM (g)	FCREm (g/g)	FCRDz (kg/dz)	LIVA (%)
0.00	101.40 ^ab^	0.9593	98.81	62.72 ^b^	60.16 ^b^	1.687 ^a^	1.270 ^ab^	100.00
0.05	103.57 ^a^	0.9638	99.27	* 64.37 ^a^	* 62.03 ^a^	1.671 ^a^	1.291 ^a^	97.38
0.10	103.08 ^a^	0.9609	98.97	* 64.96 ^a^	* 62.41 ^a^	1.652 ^ab^	1.287 ^a^	100.00
0.30	102.40 ^a^	0.9583	98.70	* 64.61 ^a^	* 61.91 ^a^	1.658 ^ab^	1.286 ^ab^	100.00
0.60	102.45 ^ab^	0.9483	97.68	* 64.83 ^a^	61.46 ^ab^	1.668 ^a^	1.297 ^a^	99.88
1.00	99.81 ^b^	0.9621	98.76	* 64.78 ^a^	* 62.32 ^a^	* 1.603 ^b^	1.246 ^b^	98.29
*p value*	0.002	0.548	0.604	<0.001	0.002	0.003	0.008	0.227
L	0.002	0.262	0.214	<0.001	0.091	<0.001	0.041	0.214
Q	0.018	0.391	0.315	0.002	0.072	0.181	0.002	0.381
SEM	4.11	0.081	4.52	6.91	3.88	0.81	0.062	8.81
CV (%)	1.71	1.79	1.81	1.09	1.82	2.29	2.10	2.68

* Means differ from control (0.0) by Dunnet’s test (*p* < 0.05). Different letters indicate statistical differences by the Tukey test (*p* < 0.05). L—linear effect; Q—quadratic effect.

**Table 3 animals-15-01318-t003:** Effect of trehalose levels on the relative weight of shell (RWS, %), yolk (RWY, %), and albumen (RWA, %), yolk color (YC), Haught Unit (HU), specific gravity (SG, g/cm^3^), thickness of shell (Tshell, μm), and shell strength (SS) of light hens eggs.

	Parameters							
Levels (%)	RWS (%)	RWY (%)	RWA (%)	YC	HU	SG (g/cm^3^)	Tshell (μm)	SS (kgf)
0.00	9.94	27.08	62.78	5.61 ^b^	83.01	1.087	0.442	3.761
0.05	9.95	27.55	62.49	5.75 ^ab^	82.36	1.087	0.449	3.518
0.10	9.94	26.99	63.07	* 5.93 ^a^	82.23	1.088	0.448	3.514
0.30	9.91	27.02	63.09	* 5.89 ^a^	81.26	1.088	0.448	3.519
0.60	9.82	26.91	63.27	5.60 ^b^	82.35	1.087	0.442	3.559
1.00	9.91	27.14	62.97	5.60 ^b^	81.83	1.088	0.445	3.711
*p-value*	0.575	0.365	0.122	<0.001	0.088	0.542	0.114	0.725
L	0.478	0.412	0.219	0.003	0.092	0.172	0.274	0.616
Q	0.512	0.331	0.264	0.008	0.081	0.476	0.382	0.812
SEM	1.21	3.14	4.73	2.38	4.61	0.021	0.008	0.122
CV (%)	1.55	2.25	0.90	2.60	1.36	0.11	1.56	9.14

* Means differ from control (0.0) by Dunnet’s test (*p* < 0.05). Different letters indicate statistical differences by the Tukey test (*p* < 0.05). L—linear effect; Q—quadratic effect.

**Table 4 animals-15-01318-t004:** Effect of trehalose levels on MDA levels (nmol/mg protein) of eggs of light hens.

Levels	Subperiods				
(%)	1	2	3	4	5
0	0.122	0.123	0.124	0.131	0.135 ^b^
0.05	0.124	0.121	0.132	0.134	0.145 ^b^
0.1	0.131	0.134	0.142	0.144	0.143 ^b^
0.3	0.114	0.122	0.132	0.154	0.211 ^a,^*
0.6	0.124	0.131	0.136	0.156	0.262 ^a,^*
1.0	0.122	0.132	0.145	0.161	0.153 ^b^
*p-value*	0.735	0.720	0.804	0.214	0.006
L	0.812	0.655	0.712	0.132	0.004
Q	0.732	0.781	0.733	0.312	0.012
SEM	0.081	0.051	0.073	0.081	0.092
CV (%)	6.87	4.36	5.43	7.62	5.62

* Means differ from control (0.0) by Dunnet’s test (*p* < 0.05). Different letters indicate statistical differences by the Tukey test (*p* < 0.05). L—linear effect; Q—quadratic effect.

**Table 5 animals-15-01318-t005:** Effect of trehalose levels on weight (TW), length (L), bone strength (BS), dry matter (DM), phosphorus (P) and calcium (Ca) of tibiotarsus of light hens.

Levels (%)	Parameters					
	TW (g)	L (mm)	BS (kgf)	DM (g/kg)	P (g/kg)	Ca (g/kg)
0.00	8.50 ^b^	115.83	20.57 ^c^	462.3	101.1	170.2
0.05	9.00 ^ab^	117.19	21.88 ^bc^	459.2	103.4	167.2
0.10	8.64 ^ab^	116.77	* 24.79 ^a^	451.2	99.3	168.4
0.30	* 9.14 ^a^	117.97	21.08 ^c^	454.7	100.1	169.1
0.60	8.62 ^ab^	116.78	19.89 ^c^	461.4	102.2	162.2
1.00	8.77 ^ab^	116.21	* 23.96 ^ab^	458.9	103.1	161.3
*p value*	0.024	0.284	<0.001	0.6402	0.890	0.596
L	0.62	0.73	0.61	0.82	0.65	0.10
Q	0.10	0.04	0.75	0.54	0.56	0.76
SEM	0.032	4.23	2.54	10.61	3.81	5.92
CV (%)	11.27	5.04	1.58	0.009	0.016	0.022

* Means differ from control (0.0) by Dunnet’s test (*p* < 0.05). Different letters indicate statistical differences by the Tukey test (*p* < 0.05). L—linear effect; Q—quadratic effect.

## Data Availability

The data are available upon request from the authors.

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
