# Peer review of "The Impact of Adding Trehalose to the Diet on Egg Quality and Tibia Strength in Light-Laying Hens"

_animals, 2025, doi:10.3390/ani15091318_

Round 1
Reviewer 1 Report
Comments and Suggestions for Authors
The authors evaluated the potential of using trehalose as a feed additive in the diets of laying hens. The study is interesting. However, the authors should revise the paper based on the reviewer's comments.
Comments:
There is a shortage in the abstract: please provide the levels of supplement used and a brief description of the results obtained with the optimal level of the supplement.
Please format the reference citations according to journal guidelines (numbered).
L75-77: How were birds fed twice, and there were ad libitum access to feed?
More information about the feed additive should be provided (source and description).
The data of all parameters measured were collected at the end of each 21 days, and then the average of the five periods was determined or what happened? Please clarify.
L133: why were the shells oven-dried if their average Wt. was calculated relative to the average egg weight?
L164: 600ºC for 4 hours, please mention the apparatus.
l193: the verb enhanced is unsuitable for the meaning.
L192: please check the levels.
L198: the results of RWS are insignificant. Rewrite.
L213-215: The antioxidant was assessed by the level of MDA. However, increasing the level of MDA is unfavorable. In addition, in the method, the authors mentioned that they assessed lipid peroxidation by measuring the TBARS.
L222: the number 0.187 refers to what?
L221-223: rephrase
Format all letters in the tables to superscript.
L231-233, 260: the decrease in FI and FCR was observed at 1%, not 0.6%.
L256, 285: format the reference to author form.
L267: probiotic?
L274-275: delete the study about humans.
L294: delete facilitated.
L299: Feed Conversion Ratio—lower case.
The discussion is poorly written and should be improved. There should be a sequence in the ideas, and they should be related to the data displayed in the results section. The authors jump from one idea to another. The results should be scientifically explained and the correlated. The authors should compare their results with previous results conducted on laying hens
L359: revise the optimal level.
Comments on the Quality of English Language
moderate editing is needed.
Author Response
Comments: There is a shortage in the abstract: please provide the levels of supplement used and a brief description of the results obtained with the optimal level of the supplement.
Response: Thank you for the comment. Adjusted as requested. The levels have been added to the abstract. Information regarding the results has been included, and the conclusion of the abstract has been adjusted in accordance with the recommended level.
Comments: L75-77: How were birds fed twice, and there was ad libitum access to feed?
Response: Adjusted as indicated. As observed, the birds were provided with two feeding opportunities, with a quantity sufficient to meet their feed intake needs, allowing ad libitum intake.
Comments: The data of all parameters measured were collected at the end of each 21 days, and then the average of the five periods was determined or what happened? Please clarify.
Response: Good suggestion, thank you. Data were collected at the end of each period, and data were obtained for each phase, with general averages calculated considering all periods.
Comments: L133: why were the shells oven-dried if their average Wt. was calculated relative to the average egg weight?
Response: The shells were oven-dried to account for the residual albumen contained within them. After weighing each shell, considering the absolute weight of the egg from which it came, the relative shell weight was determined. The adjustment was made for the albumen weight, considering the weight difference between wet (with albumen residue) and dry shells.
Comments: L164: 600ºC for 4 hours, please mention the apparatus.
Response: The model has been added as requested.
Comments: l193: the verb enhanced is unsuitable for the meaning.
Response: The text has been edited as requested.
Comments: L192: please check the levels.
Response: Levels have been checked; thank you for the observation.
Comments: Please format the reference citations according to journal guidelines (numbered).
Response: References have been formatted accordingly.
Comments: L198: the results of RWS are insignificant. Rewrite.
Response: Likely referring to line 218. Adjust.
Comments: L213-215: The antioxidant was assessed by the level of MDA. However, increasing the level of MDA is unfavorable. In addition, in the method, the authors mentioned that they assessed lipid peroxidation by measuring the TBARS.
Response: Thank you for the comment. Additional information has been included to explain that the MDA data were incorrectly reported.
Comments: More information about the feed additive should be provided (source and description).
Response: More information about the additive, Trehalose, has been added.
Comments: L222: the number 0.187 refers to what?
Response: A very good point, thank you. The information has been adjusted, as this number was incorrect. It has been rewritten.
Comments: L221-223: rephrase
Response: Rephrased as requested after adjusting the previous recommendation.
Comments: Format all letters in the tables to superscript.
Response: Done as requested.
Comments: L231-233, 260: the decrease in FI and FCR was observed at 1%, not 0.6%.
Response: Thank you for the correction. Adjusted accordingly.
Comments: L256, 285: format the reference to author form.
Response: Thank you for the correction. Adjusted accordingly.
Comments: L267: probiotic?
Response: Edited to prebiotic. Thanks for your observation.
Comments: L274-275: delete the study about humans.
Response: Edited as requested.
Comments: L294: delete facilitated.
Response: Done as requested.
Comments: L299: Feed Conversion Ratio—lower case.
Response: Done as requested.
Comments: The discussion is poorly written and should be improved. There should be a sequence in the ideas, and they should be related to the data displayed in the results section. The authors jump from one idea to another. The results should be scientifically explained and correlated. The authors should compare their results with previous results conducted on laying hens.
Response: The discussion has been completely rewritten, adjusting and addressing the reviewers' suggestions.
Comments: L359: revise the optimal level.
Response: Revised as requested.

Reviewer 2 Report
Comments and Suggestions for Authors
The information could be useful, but the scientific presentation quality needs to improved.
Here are some suggestions:
1) Abstract must be more descriptive for the results.
2) Introduction is poor with only two references cited here.
3) Materials and method sections should be subdivided in subsections.
4) footnote to the tables: delete any regressions equations used. They are not useful.
5) Tables 2 and onward: Present SEM values instead of CV.
6) All tables from Table 2: instead of ns, present actual p-values. P-values may be maximum 3 decimal places.
7) Try to reduce decimal places for the data presented in tables.
8) L238-259: It is like a review paper. It may be shifted to Introduction, which was weak.
9) Discussion should be subdivided like results. In the present context, most of the texts looks like a review paper. Authors should discuss the results in the present study.
Comments on the Quality of English Language
Needs revisions for improvement of English expressions.
Author Response
Comments: Abstract must be more descriptive for the results.
Response: Done as requested.
Comments: Introduction is poor with only two references cited here.
Response: The introduction has been revised to include additional references, particularly focusing on poultry, to emphasize the significance of this study with laying hens.
Comments: 3) Materials and method sections should be subdivided in subsections.
Response: Adjusted as requested.
Comments: 4) Footnote to the tables: delete any regression equations used. They are not useful.
Response: The regression equations have been retained, as they cite data determined for the additive. While we understand they might be removed, we believe they provide valuable information to the material.
Comments: 5) Tables 2 and onward: Present SEM values instead of CV.
Response: SEM values have been added to the tables.
Comments: 6) All tables from Table 2: instead of ns, present actual p-values. P-values may be a maximum of 3 decimal places.
Response: Adjusted as requested.
Comments: 7) Try to reduce decimal places for the data presented in tables.
Response: Done as requested.
Comments: 8) L238-259: It is like a review paper. It may be shifted to Introduction, which was weak.
Response: The section has been adjusted and moved to the Introduction for better contextualization.
Comments: 9) Discussion should be subdivided like results. In the present context, most of the text looks like a review paper. Authors should discuss the results in the present study.
Response: The Discussion has been revised to focus more on the study's results and subdivided as requested for better clarity.
Comments: Needs revisions for improvement of English expressions.
Response: Revisions have been made to improve English expressions.
Round 2
Reviewer 1 Report
Comments and Suggestions for Authors
Thank you for the revision
Minor comments:
Line 132: your production?
Line 250-254: delete numbers in the text.
Comments on the Quality of English LanguageModerate
Author Response
Dear Reviewer,
We sincerely appreciate your thorough review and constructive feedback, which have been invaluable in improving the quality of our manuscript. We have taken your recommendations into account and made meticulous revisions to both the grammar and content throughout the paper. Our focus was primarily on the contextualization and discussion sections.
We have carefully implemented all of the requested changes to enhance the clarity and scientific rigor of our work. Additionally, we have reviewed the entire manuscript to ensure the highest overall quality.
----
Minor comments:
Comment: Line 132: your production?
Response: It has been adjusted, thank you for your review.
Comment: Line 250-254: delete numbers in the text.
Response: It has been adjusted, thank you for your observation. I believe it was due to text adjustments, which copied the line numbers.
----
Thank you once again for your time and effort in helping us refine our work.
Sincerely,
Matheus

Reviewer 2 Report
Comments and Suggestions for Authors
Authors have considered most of the suggestions except the followings:
1) I strongly suggest to edit this paper with English editing service. It was also stated in the first review.
2) Is there any justified reason that you d o not want remove the regression equations from the footnote of the tables. You have already showed p-values of polynomial contrasts - then regression equations are needed. Are you predicting here and it is the objective of the study?
3) Comments: 9) Discussion should be subdivided like results. In the present context, most of the text looks like a review paper. Authors should discuss the results in the present study.
Response: The Discussion has been revised to focus more on the study's results and subdivided as requested for better clarity.
Comments: you have mentioned you have subdivided the discussion, but you have not.
Other comments
footnote to the table: delete ns as now it is not there.
L310-328: Not clear why the results are stated here mostly. Also, it seems here the summary of the all results and conclusion. It may be okay in conclusion, but not here.
L330-342: Statements should be supported from the literature.
L358-370: There is not citations - please cite some references when you mention a study. What was the study?
L421-430: Not clear why are talking much here. Maybe one sentence will be fine here or delete it.
L431-443: Provide some citations here to justify your statements - presently there is not citation.
L445-448: "Conversely, when one considers the significant and multifaceted role that trehalose plays as a vital energy reservoir within the intricate biological systems of various animal species, it becomes apparent that the supplementation of this sugar may have induced notable and measurable alterations in the overall feed efficiency observed in these organisms." This is a verbose sentence. Delete it.
L497-507: This is not a discussion - but summary, which is not needed here. You have already stated this in the conclusion section.
Overall, discussion is poor without proper citation in scientific discussion. English needs improvement.
Comments on the Quality of English LanguageIt must be edited, which was also indicated in the previous version.
Author Response
Dear Reviewer,
We sincerely appreciate your thorough review and constructive feedback, which have been invaluable in improving the quality of our manuscript. We have taken your recommendations into account and made meticulous revisions to both the grammar and content throughout the paper. Our focus was primarily on the contextualization and discussion sections.
We have carefully implemented all of the requested changes to enhance the clarity and scientific rigor of our work. Additionally, we have reviewed the entire manuscript to ensure the highest overall quality.
----
Comment: 1) I strongly suggest to edit this paper with English editing service. It was also stated in the first review.
Response: Thank you for your continued support. In this latest version, I received assistance from a native English-speaking colleague from our university, which led to improvements in the language.
Comment: 2) Is there any justified reason that you do not want remove the regression equations from the footnote of the tables. You have already showed p-values of polynomial contrasts - then regression equations are needed. Are you predicting here and it is the objective of the study?
Response: I agree with your point. We have adjusted this section and removed the polynomial regression equations as suggested.
Comment: 3) Comments: 9) Discussion should be subdivided like results. In the present context, most of the text looks like a review paper. Authors should discuss the results in the present study.
Response: The Discussion has been revised to focus more on the study's results and subdivided as requested for better clarity. Comments: you have mentioned you have subdivided the discussion, but you have not.
Response: You are correct, and I appreciate your observation. In this latest version, we have addressed this issue by subdividing the discussion into clear subsections for better comprehension. Additionally, we conducted a thorough revision of both the language and the contextual presentation.
Other comments
Comment: Footnote to the table: delete "ns" as it is no longer present.
Response: Absolutely correct. I had forgotten to remove it, and I appreciate the reminder.
Comment: L310-328: It is unclear why most of the results are presented here. This section seems to summarize all the results and conclusions, which may be appropriate for the conclusion but not here.
Response: This section has been revised as requested.
Comment: L330-342: Statements should be supported by literature references.
Response: The statements have been adjusted, and references have been added as requested.
Comment: L358-370: There are no citations provided. Please include references when discussing a study. Which study are you referring to?
Response: Citations have been added as requested.
Comment: L421-430: The text here seems unnecessarily lengthy. One sentence may suffice, or consider deleting this section.
Response: The text has been condensed as requested.
Comment: L431-443: Please provide citations to support your statements, as none are currently provided.
Response: Citations have been added to support the statements as requested.
Comment: L445-448: "Conversely, when one considers the significant and multifaceted role that trehalose plays as a vital energy reservoir within the intricate biological systems of various animal species, it becomes apparent that the supplementation of this sugar may have induced notable and measurable alterations in the overall feed efficiency observed in these organisms." This sentence is overly verbose. Please delete it.
Response: The sentence has been deleted as requested.
Comment: L497-507: This is not a discussion but rather a summary, which is unnecessary here since it has already been presented in the conclusion section.
Response: This section has been revised to align with your suggestion.
Comment: Overall, the discussion is lacking proper citations for a scientific discussion. Additionally, the English needs improvement.
Response: The discussion has been enriched with the appropriate citations, and improvements have been made to the English as suggested.
----
Thank you once again for your time and effort in helping us refine our work.
Sincerely,
Matheus
